# Improving Retrieval in Theme-specific Applications using a Corpus Topical Taxonomy

## ABSTRACT

Document retrieval has greatly benefited from the advancements of large-scale pre-trained language models (PLMs) owing to their superior capability to understand contextual semantics. However, their effectiveness is often limited in theme-specific applications for specialized areas or industries, due to unique terminologies, incomplete contexts of user queries, and specialized search intents. To capture the theme-specific information and improve retrieval, we propose to use a *corpus topical taxonomy*, which outlines the latent topic structure of the corpus while reflecting user-interested aspects. We introduce ToTER (Topical Taxonomy Enhanced Retrieval) framework, which identifies the central topics of queries and documents with the guidance of the taxonomy, and exploits their topical relatedness to supplement missing contexts. As a plug-and-play framework, ToTER can be flexibly employed to enhance various PLM-based retrievers. Through extensive quantitative, ablative, and exploratory experiments on two real-world datasets, we ascertain the benefits of using topical taxonomy for retrieval in theme-specific applications and demonstrate the effectiveness of ToTER.

## CCS CONCEPTS

• **Information systems** → **Information retrieval**; *Specialized information retrieval*; *Document topic models*.

## KEYWORDS

Document retrieval, Topical taxonomy, Theme-specific application

**ACM Reference Format:**
Anonymous Author(s). 2018. Improving Retrieval in Theme-specific Applications using a Corpus Topical Taxonomy. In *Proceedings of ACM Web Conference (Woodstock '18)*. ACM, New York, NY, USA, 14 pages.

## 1 INTRODUCTION

Pre-trained language models (PLMs) have improved document retrieval owing to their superior semantic understanding [17, 21, 22, 58]. The PLM-based retrieval models are first pre-trained on the massive textual corpora to grasp language understanding. Subsequently, they are fine-tuned using vast datasets of annotated query-document pairs, which enables the models to capture their semantic similarities within the latent space. While successful in general domains like web search which consist of a broad user base, they are often limited in *specialized applications with specific themes*.

Theme-specific applications are specialized areas or industries where retrieval tasks are focused on a specific theme (e.g., academic paper search, product search in e-commerce). Retrieval in theme-specific applications poses three challenges spanning specialized terminology and niche content (C1), limited contexts of user query (C2), and specialized user interests and search intents (C3).

**C1**: Theme-specific domains often have specialized terminologies, which are not frequently included in the general text corpus. For example, Table 1(a) shows that an academic paper includes many technical terms specific to certain research fields, such as "proof of retrievability" and "cryptographic proof". PLM-based retrievers trained on general text corpora often lack an inherent understanding of domain-specific specialized and niche terminologies [8].

**C2**: Users familiar with the domain often omit contexts they believe are naturally implied in their query. For example, in product search, users enter a query such as "RTX 3090" without adding contexts such as "graphics cards". Table 1(a) shows queries from domain experts may skip over general contexts such as "cryptography" or "computer security". Omitted terms hinder the model's ability to fully comprehend the query, leading to imprecise retrieval outcomes. Inferring missing contexts is more challenging in theme-specific applications as it often requires domain-specific knowledge.

**C3**: Users in theme-specific applications have more specialized interests and intents compared to general web searches. For example, researchers may want to find papers within a specific field of study to discern a particular research trajectory. In product search, users often filter results based on specific product attributes. For example, Table 1(b) shows that both documents are somewhat relevant to the query as both of them are about ammonia-free hair color products. However, the query targets hair dye with lasting effects, instead of hair rinse with temporary effects. These specialized search intents are not effectively captured by models trained on general corpora.

Accumulating ample labeled data can mitigate these challenges to some extent. However, the creation of such datasets in theme-specific applications is particularly challenging due to the need for domain expertise (e.g., academic domain) and the proprietary nature of user logs in specialized applications (e.g., e-commerce) [4, 30]. As a result, PLM-based retrieval models often struggle to accurately capture relevance in theme-specific applications [52].

To improve retrieval without relying on labeled data, we propose to use a *corpus topical taxonomy* [15, 28, 39, 50, 61], which has been extensively studied for organizing topics in a corpus. A corpus topical taxonomy outlines the latent topic hierarchy within the corpus as a tree structure, where each node is a *topic class* represented by a cluster of semantically coherent terms describing the topic, as shown in Figure 1. Recent taxonomy construction studies [1, 15, 28] have effectively reflected user-interested aspects, drawing from a foundational seed taxonomy rooted in human knowledge of the application (e.g., fields of study from Mircosoft Academic [51]). The constructed taxonomy can be subsequently employed to provide

**Table 1: Examples of retrieval in theme-specific applications. We use Contriever-MS (retriever) and MiniLM-L-12 (reranker). Contents closely related to the query are denoted in bold. Details of topic class and core phrase discovery are provided in §4.**

| (a) Academic domain | | (b) Product domain | |
| --- | --- | --- | --- |
| Query | Provable data possession at untrusted stores | Query | #1 black natural hair dye without ammonia or peroxide |
| Document A (relevant, rank: top-173) | Pors: **proofs of retrievability** for large files. In this paper, we define and explore proofs of retrievability (PORs). ... A POR may be viewed as a kind of **cryptographic proof** of knowledge (POK). ... We view PORs as an important tool for **semi-trusted** online archives. Existing cryptographic techniques help users **ensure the privacy and integrity** of files they retrieve. ... | Document A (relevant, rank: top-70) | ONC NATURALCOLORS (1N **Black**) 4 fl. oz. (120 mL). Healthier permanent **hair dye** with certified organic ingredients, **ammonia free**, vegan friendly, 100 gray coverage. ... Cruelty-free and vegan. It is time to make the clean choice. |
| | | Document B (irrelevant, rank: top-11) | Roux Fanci-full Rinse 16 Hidden Honey. Tones and enhances gray and blonde hair. Rinses in and shampoos out. **No ammonia or peroxide**. ... 15 applications per bottle, temporary **hair color**, 15 ounce bottle. |
| ToTER rank: top-10 | Topic classes: cryptography, trusted computing, digital content, computer network, computer security, computer science | ToTER rank: top-5 (Doc.A) top-32 (Doc.B) | Topic classes: hair color, hair coloring products, hair care, beauty & personal care |
| | Core phrases: encryption, access control, security, key, server | | Core phrases: dye, permanent, lasting, permanent hair color, ammonia free |

additional clues to link queries and documents by discerning their topical relatedness and supplementing the missing contexts. However, the potential of such taxonomies in enhancing PLM-based retrieval remains unexplored in the previous literature.

We propose **To**pical **T**axonomy **E**nhanced **R**etrieval (ToTER) framework, which systematically leverages the corpus taxonomy to complement the semantic matching of PLM-based retrieval. The taxonomy provides a high-level topic hierarchy of the entire corpus. To harness this corpus-level knowledge for retrieval, we first link it to individual documents. Specifically, ToTER first conducts *topic class relevance learning* to discern the relevance of each document to each topic class node in the taxonomy. We formulate this step as an unsupervised multi-label classification problem without document-topic labels. ToTER introduces a new silver label generation strategy along with a new collective distillation process to produce rich and reliable signals. This class relevance learning allows ToTER to effectively identify central subjects of a given text under the guidance of the topical taxonomy reflecting user interests.

Based on the identified topic class relevance, ToTER leverages the topical relatedness of a query and documents to complement the semantic matching by PLM-based retrievers. In Table 1(a), we see that ToTER can improve retrieval by identifying common topic classes like "cryptography" and "computer security" for both query and document, given the presence of terms frequently used for these topic classes (**C1**). Furthermore, ToTER combines the topical relatedness with more fine-grained phrase knowledge for each topic class, helping to distinguish documents having similar topics. In Table 1(b), ToTER identifies and utilizes core topical phrases such as "dye", "lasting", and "permanent hair color" to enrich the query, enabling more accurate finding of relevant documents (**C2**). This entire process is built upon the topical taxonomy reflecting user-interested aspects (**C3**). Formally, ToTER introduces three strategies to complement the PLM-based retrieval: (1) search space adjustment, (2) class relevance matching, and (3) query enrichment by core phrases. Our contributions are summarized as follows:

- We present a systematic approach to incorporate topical taxonomy into retrieval in theme-specific applications, which is new for PLM-based retrieval.
- We propose ToTER that deliberately discerns and utilizes topical relatedness of queries and documents. As a plug-and-play framework, it can be integrated with various PLM-based models.

- We validate the effectiveness of ToTER by extensive experiments. ToTER consistently improves retrieval accuracy in scenarios with both no labeled data and limited labels. Furthermore, we provide an in-depth analysis of the ToTER framework.

## 2 PROBLEM FORMULATION

### 2.1 Concept Definition

**PLM-based multi-stage retrieval.** Most PLM-based retrieval systems leverage multi-stage retrieve-then-rerank pipeline [8, 33, 62]. Specifically, given a query $q$, a *retriever* retrieves a set of candidate documents $\mathcal{D}_q$ from a large corpus $\mathcal{D}$, where $|\mathcal{D}_q| \ll |\mathcal{D}|$. Following the first-stage retrieval, a *reranker* computes a more fine-grained relevance for each candidate document, and generates the final ranked list by reordering them.

The first-stage retriever typically adopts a dual-encoder architecture, where query and documents are encoded separately and the relevance is measured by the similarity between their embeddings.

$$s_{de}(q, d) = sim(f_\theta(q), f_{\theta'}(d)), \tag{1}$$

where $f_\theta$ and $f_{\theta'}$ are the query and document encoders, and $sim(\cdot, \cdot)$ is the similarity function such as the cosine similarity. The document embeddings are pre-computed and efficiently retrieved via approximate nearest-neighbor (ANN) search techniques [20].

The second-stage reranker mostly adopts a cross-encoder architecture which takes the concatenation of a query and a document as input and assesses its relevance score.

$$s_{ce}(q, d) = f_\phi(q, d), \tag{2}$$

where $f_\phi$ denotes the reranker. By fully exploring the interactions between the query and document, it generates more accurate relevance scores compared to the dual-encoder [62].

**Topical taxonomy.** A *corpus topical taxonomy* $\mathcal{T} = (C, \mathcal{R})$ represents a tree structure that outlines the latent topic hierarchy within the target corpus. Each node $c_j \in C$ corresponds to a *topic class* which is described by a coherent cluster of terms[1] describing the topic, denoted by $P_j$. The most salient term (i.e., center term) is utilized as the class name. Each edge ($\in \mathcal{R}$) indicates a theme-specific

---

[1]Each term is regarded as a phrase composed of one or multiple word tokens, so the terms "phrase" and "term" are used interchangeably in this paper.

relationship between a parent and child node, such as "is a subfield of" or "is a type of". Figure 1 shows an example of topical taxonomy.

To construct the topical structure reflecting theme-specific user interests, taxonomies are built upon a foundational seed taxonomy rooted in human knowledge of the application [1, 15, 28]. An example is the fields of study in the academic domain [51], which embodies researchers' inclination to organize academic concepts and studies. Based on this application knowledge, recent methods [15, 28, 29, 64] have effectively generated taxonomy having remarkable term coherency, topic coverage, and user-interest alignment.

## 2.2 Problem Definition

We focus on retrieval within theme-specific applications, which are systems tailored for particular fields or industries. Given a target corpus $\mathcal{D}$ and its topical taxonomy $\mathcal{T}^2$, our goal is to develop a systematic framework that exploits the topic hierarchy knowledge to improve the existing PLM-based multi-stage retrieval. We pursue a solution that can be flexibly integrated with various existing retriever and reranker models, while effectively providing complementary knowledge for each retrieval stage.

## 3 RELATED WORK

**PLM-based retrieval models.** PLM-based retrieval models have advanced in both training and encoding strategies. In terms of training strategy, starting from in-batch and hard negatives by BM25 [21, 32], advanced hard negative mining by dynamic sampling [60] and denoising using a cross-encoder [44] have been studied. Many works have focused on pre-training with unsupervised contrastive learning [11, 12, 17], knowledge graph [8, 23], and synthetic data [4, 6, 33] to improve the capability of models. Recent studies [48, 62] also show that joint training of the retriever and reranker can further improve their effectiveness. In terms of encoding strategy, single-vector representation models [58] encode a given text as a single vector, and multi-vector representation models [22] use multiple vectors to improve expressiveness. Recent sparse representation models [9, 10] use sparse lexical representations based on the logits of the masked language model layer of PLMs, which enables a natural query and document expansion.

Despite their effectiveness, they require fine-tuning with massive labeled data to be adapted to the new domain corpus. As a plug-and-play framework, ToTER complements the above approaches using a topical taxonomy without resorting to the labeled data.

**Retrieval with auxiliary corpus knowledge.** These techniques aim to improve retrieval by exploiting knowledge of the target corpus. One notable approach is pseudo-relevance feedback (PRF), which utilizes the top-ranked results from an initial retrieval to enhance the semantic matching process. The existing methods have exploited key terms [24, 55], text segments [65], and documents [59] from top-ranked results as an additional context to complement the query. Recently, [35, 37] have directly utilized knowledge stored in the PLMs for query expansion. Despite their effectiveness in filling missing contexts, their effectiveness is often limited in theme-specific applications due to the suboptimal initial retrieval quality and the need for domain-specific knowledge.

Another approach leverages inter-document similarity via a *corpus graph* whose nodes are documents and edges connect most similar documents. Under the assumption that similar documents tend to be relevant to the same query, [34] adapts the candidate document set for reranker using the nearest neighbors in the graph. [26] first uses lexical retrieval to obtain seed documents and uses the graph to gradually expand the search space for retrievers. Lastly, there have been a few attempts to use topic information for retrieval. [19, 56] combine LDA [3] with statistic-based retrieval like query-likelihood, and [14, 53] use topics for a balanced batch construction. [31] incorporates topic information into a word embedding-based model. However, there has been no attempt to exploit the high-quality corpus taxonomy, and most methods leveraging topic information [19, 31, 56] are tailored for traditional retrieval models (e.g., statistic-based) and are difficult to apply to PLM-based models.

It is worth noting that there exist a few attempts to use external knowledge (e.g., knowledge base) [8]. We focus on exploiting knowledge of the target corpus, and ToTER can be combined with the external knowledge-based models as well. We provide related work for topic mining and taxonomy completion in Appendix A.1.

## 4 METHODOLOGY

We present **To**pical **Ta**xonomy-**E**nhanced **R**etrieval (ToTER) framework. We first explain how ToTER bridges the given taxonomy with the target corpus in the training phase (§4.1), then present how ToTER enhances PLM-based retrieval in the inference phase (§4.2). ToTER does not require labeled data from the target corpus, but it can also effectively leverage the labeled data (§4.3). The overview of ToTER is presented in Figure 1.

## 4.1 Topic Class Relevance Learning

The topical taxonomy reveals the latent structure of the whole corpus. To exploit it for document retrieval, we first connect the corpus-level knowledge to individual documents. We formulate this step as an unsupervised *multi-label classification*, assessing the relevance of each document to each topic class without document-topic labels. We first introduce our silver label generation strategy (§4.1.1). Then, we propose a new training strategy to produce rich and reliable training signals (§4.1.2). The overall training process is provided in Appendix A.2.

*4.1.1 Taxonomy-guided silver label generation.* As the first step, we seek to identify a small candidate set of relevant classes for each document, which will be our silver labels for training. Utilizing the hierarchical structure of topic classes, we introduce a top-down approach that *recursively assigns* documents to the child node with the highest similarity, gradually narrowing down the topic. Given a document, we start from the "root node" and compute its similarity to each child node. The document is then assigned to the child node with the highest similarity.[3] This assignment process recurs until it reaches leaf nodes. Once every document has been assigned, we apply a *filtering step* to retain only reliable labels for each document.

**Document-class similarity computation.** For each document $d$, we compute its similarity with a child node $c_j$ by considering all

---

[2]The topical taxonomy is obtained using the existing taxonomy completion approach [28]. We provide details of the taxonomy construction in Appendix A.3.2.

[3]Although a document can cover multiple topics, we assign it to the most similar child node to generate reliable silver labels. Indeed, we observe increasing the assignment number easily leads to degraded training efficacy.

**Figure 1: The overview of Topical Taxonomy-Enhanced Retrieval (ToTER) framework.**

phrases related to $c_j$. Let $P_j^{\mathcal{T}}$ denote the union of all phrases from the subtree having $c_j$ as a root node. The similarity is computed by considering both lexical aspect ($sim_L$) based on statistics and semantic aspect ($sim_S$) based on PLMs, which are defined as:

$$sim_L(d, c_j) = \frac{1}{|P_j^{\mathcal{T}}|} \sum_{p \in P_j^{\mathcal{T}}} \mathrm{tf}(p,d) \cdot \mathrm{idf}(p)$$

$$sim_S(d, c_j) = \frac{1}{|P_j^{\mathcal{T}}|} \sum_{p \in P_j^{\mathcal{T}}} \cos(\mathbf{h}_p, \mathbf{h}_d), \quad (3)$$

where $\mathbf{h}_p$ and $\mathbf{h}_d$ denote representations from PLM for a phrase $p$ and a document $d$.[4] The lexical and semantic similarity can reveal complementary aspects; lexical matching has strengths in handling domain-specific terminologies that rarely exist in the general corpus, and semantic matching excels in capturing broader contextual meanings while flexibly handling non-exact matching terms.

To jointly consider the two aspects, we adopt the ensemble score based on the reciprocal rank [63, 64]. By ranking all child nodes in descending order of $sim_L$ and $sim_S$, each node will have two rank positions $rank_L$ and $rank_S$, respectively. The overall similarity is:

$$sim_O(d, c_j) = \left( \frac{1}{2} \left( \frac{1}{rank_L(c_j)} \right)^\rho + \frac{1}{2} \left( \frac{1}{rank_S(c_j)} \right)^\rho \right)^{1/\rho}, \quad (4)$$

where $0 < \rho \le 1$ is a constant. We set $\rho = 0.1$.

**Filtering step.** After the class assignment for all documents, we apply a filtering step to only retain assignments with high similarity. For each class node, we keep documents whose similarity exceeds the median similarity of all documents assigned to the class. If a document is filtered out from a certain node, the document is also removed from all its child nodes, ensuring hierarchical consistency.[5] Finally, for each document $d$, we obtain silver labels $\mathbf{y}_d^s \in \{0, 1\}^{|C|}$, where $y_{dj}^s = 1$ if $c_j$ is assigned to $d$, otherwise 0.

---

[4]We employ BERT [7] and apply mean pooling to obtain the final representation.
[5]We apply the filtering to nodes at the second or deeper levels to ensure every document has at least one class assignment (the root node has level 0).

*4.1.2 Class relevance learning.* Based on the obtained silver labels, we train a *class relevance estimator* that predicts relevance between a document and topic classes. For effective training, we propose a new *collective topic knowledge distillation* strategy designed to complement incomplete silver labels.

**Class relevance estimator.** As developing a novel architecture is not the focus of this paper, we employ the existing methods to encode documents and class nodes. For the *text encoder*, we use BERT [7] with mean pooling to obtain $\mathbf{h}_d$ for each document $d$. For the *topic class encoder*, we adopt graph convolutional networks (GCNs) [25] to incorporate both semantic and structural information. For each class node $c_j$, we first obtain its ego graph that includes its $L$-hop neighboring nodes and apply GCN layers to propagate node features over the taxonomy structure. Each node feature is initialized by the BERT representation of its class name. After stacking $L$ GCN layers, we use the representation of the ego node, denoted as $\mathbf{c}_j$, as the final class representation.

Then, we calculate the topic class-document relevance by bilinear interaction between their representations, i.e., $\hat{y}_{dj} = \sigma(\mathbf{c}_j^\top \mathbf{M} \mathbf{h}_d)$, where $\mathbf{M}$ is a trainable interaction matrix and $\sigma(\cdot)$ is the sigmoid function. The estimator is trained by the binary cross-entropy loss:

$$\min_{\mathbf{M}, \theta_{GCN}} \mathcal{L} = - \sum_{d \in \mathcal{D}} \sum_{c_j \in C} y_{dj} \log \hat{y}_{dj} + (1 - y_{dj}) \log(1 - \hat{y}_{dj}). \quad (5)$$

In the early stages of training, we use the silver labels $\mathbf{y}_d = \mathbf{y}_d^s$. As training progresses, we exploit collective topic labels $\mathbf{y}_d = \mathbf{y}_d^c$ obtained from similar documents, which we will introduce below.

**Collective topic knowledge distillation (CKD).** As the silver labels incompletely reveal true relevance classes, relying solely on them leads to suboptimal estimation. As a solution, we propose CKD, designed to complement the incomplete labels. Our core idea is that the topic distribution of a document can be inferred from semantically similar documents. Specifically, (1) Using each document $d$ as a query, we *retrieve* a small subset of semantically similar documents $\mathcal{D}_d$ from the corpus. To accurately retrieve $\mathcal{D}_d$, we

consider both semantic similarity (via a dual-encoder) and topical relatedness (via our estimator). This will be explained in Sec 4.2.2. (2) We compute the class relevance distributions for each retrieved document, i.e., $\{\hat{\mathbf{y}}_{d'} \mid d' \in \mathcal{D}_d\}$. (3) By averaging the predicted distributions, we generate collective relevance labels $\mathbf{y}_d^c \in (0, 1)^{|C|}$.

Unlike $\mathbf{y}_d^s$ which consists of binary values, $\mathbf{y}_d^c$ reveals the soft probability of a document's relevance to each class. Notably, $\mathbf{y}_d^c$ reveals topic classes that are highly pertinent to documents similar to $d$, providing rich supervision not included from $\mathbf{y}_d^s$. Moreover, this collective knowledge distills more stable and reliable signals than using individual predictions for pseudo-labeling, as done in conventional self-training [16, 57]. It is worth noting that $\mathbf{y}_d^c$ gets refined during the training. That is, the topic estimator is improved with collective knowledge, which again results in more accurate discovery of similar documents and their topic distributions.

## 4.2 Topical Taxonomy-Enhanced Retrieval

We present how ToTER improves PLM-based retrieval at the inference phase. ToTER consists of three strategies to complement the existing retrieve-then-rerank pipeline. Each strategy is designed to gradually focus on fine-grained ranking, as shown in Figure 1.

**Class relevance estimation.** After training, for every document $d$ in the corpus, we compute its topic class relevance as $\hat{\mathbf{y}}_d$. Considering each document only covers a small subset of topics within the corpus, we focus on classes with high relevance. To indicate these *relevant classes*, we introduce a binary indicator vector $\hat{\mathbf{b}}_d \in \{0, 1\}^{|C|}$, where $\hat{b}_{dj} = 1$ denotes that $d$ has a certain degree of relevance to $c_j$, otherwise $\hat{b}_{dj} = 0$. We recursively retain the top $m\%$ classes for each level of taxonomy by setting the corresponding elements of $\hat{\mathbf{b}}_d$ as 1. If a class is not retained at a higher level, all its child classes are not retained as well, ensuring hierarchical consistency. We set $m = 10$. At test time, for a given query $q$, we obtain $\hat{\mathbf{y}}_q$ and $\hat{\mathbf{b}}_q$ in the same way.

### 4.2.1 Search space adjustment (SSA) to reduce initial search space.
The topic class relevance reveals the central subjects of each document, providing a snapshot of its main focus. Before applying the PLM-based retrievers, we seek to *filter out* a large number of irrelevant documents having little topic class overlap with the query. This step can benefit subsequent retrieval by reducing the search space while preserving topically relevant documents that may otherwise be overlooked by PLM-based retrievers. For search space reduction, lexical models (e.g., BM25) are mostly considered due to their high efficiency [26, 52]. We expect topic-based SSA can have strengths in identifying relevant documents, compared to using lexical similarity based on word overlap.

As topics are discrete categories, we can efficiently compute the topic overlap using the binary vectors. In specific, we compute the degree of topic overlap between the query and each document using *bitwise operations*: Popcount(AND($\hat{\mathbf{b}}_q, \hat{\mathbf{b}}_d$)). Then, we filter out documents with low degrees of overlap, obtaining the reduced search space $\mathcal{D}_q^{SSA}$. The size of this space can be determined empirically. We continue subsequent retrieval on $\mathcal{D}_q^{SSA}$ instead of $\mathcal{D}$.

**Remarks.** Compared to using real-valued vectors $\hat{\mathbf{y}}_*$, the proposed SSA is more efficient as it uses bitwise operations of binary vectors, largely reducing the need for floating-point operations. It

can be further accelerated using multi-index hashing for binary codes [43]. As search speed acceleration is a distinct research topic, we focus on the accuracy aspect in this work.

### 4.2.2 Class relevance matching (CRM) for retriever.
The first-stage retriever aims to find a set of candidate documents $\mathcal{D}_q$. In this step, we exploit *topical relatedness* of the query and document, which is the similarity between distributions of the relevant topic classes. Topical relatedness focuses on the relevance of the central subjects of the input texts identified using the class estimator. This can help to handle lexical mismatches and fill in missing contexts, providing a complementary aspect to semantic similarity by the dual-encoder. Formally, CRM retrieves $\mathcal{D}_q$ based on $s(q, d)$, considering both semantic similarity from dual-encoder $s_{de}(q, d)$ and topical relatedness $s_{CRM}(q, d)$:

$$s(q, d) = combine(s_{de}(q, d), s_{CRM}(q, d)),$$
$$s_{CRM}(q, d) = sim(\hat{\mathbf{y}}_q \odot \hat{\mathbf{b}}_q, \hat{\mathbf{y}}_d \odot \hat{\mathbf{b}}_d). \quad (6)$$

We obtain the relevant class distribution using element-wise multiplication, denoted as $\odot$, with the binary vector. Then, we compute the similarity between the query and document distributions using the $sim(\cdot, \cdot)$ function, where we use inner-product. In practice, we can only save and check scores of the relevant classes (i.e., $\{\hat{y}_{*j} | \hat{b}_{*j} = 1\}$) to improve efficiency. $combine(\cdot, \cdot)$ denotes a function to consolidate two scores, and we adopt a simple addition with rescaling via z-score normalization.[6] We also explored reflecting the granularity of topics by exclusively focusing on broad or specific topic classes in CRM. However, considering the overall topic distribution, encompassing both broad and specific topics, proved most effective. Please refer to Appendix A.4.1 for a detailed study.

### 4.2.3 Query enrichment by core phrases (QEP) for reranker.
In this last stage, a reranker with cross-encoder architecture reorders $\mathcal{D}_q$ based on their fine-grained relevance to the query $q$. Since $\mathcal{D}_q$ already have similar relevant classes via CRM, in this step, we delve deeper into each topic by focusing on class-related phrases. As discussed earlier, users familiar with a domain often omit contexts in their queries, which makes it difficult to find accurate relevance. To address this, we use phrase-level knowledge to enrich queries.

A remaining question is how to identify phrases to complement a given query. QEP is built upon the relevance model philosophy [27], which assumes that both a query and its relevant documents are generated from a shared underlying relevance model. Although the true relevance model behind the query is unknown, it can be inferred from the most relevant documents obtained from retrieval [55, 59, 65]. Based on this idea, we identify *core phrases* to enrich the query using both the topic class knowledge and top-ranked retrieved documents. From the set of relevant class phrases $\{P_j | \hat{b}_{qj} = 1\}$, we collect the top-$k$ core phrases $P^q$ that most frequently appear in top-ranked documents.[7] Then, we enrich the original query with $P^q$ and use the enriched query for inference:

$$s_{ce}(q, d) = f_\phi(q^{QEP}, d), \quad q^{QEP} = [q; \text{TMPLT}(P^q)], \quad (7)$$

---

[6] A hyperparameter can be also used to balance two scores of varying magnitudes. However, we consistently achieved satisfactory results using simple normalization.
[7] The phrases related to each class ($P_j$) are provided in the topical taxonomy (§2.1). The frequencies of phrases for each document are pre-computed.

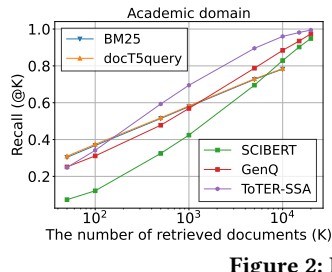
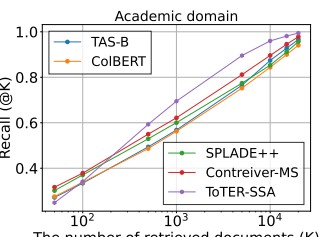
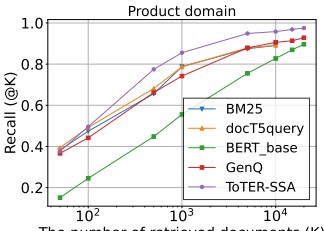
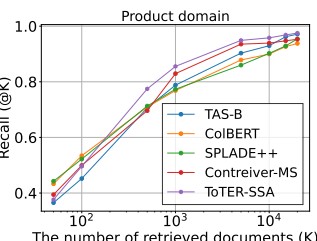

**Figure 2: Retrieval performance comparison on the academic and product domain.**

where TMPLT denotes the template of the hard prompt. In this work, we use $\text{TMPLT}(P^q) = $ ", *relevant topics:* $\{P^q\}$". We also tried using just the topic class names (i.e., the most salient phrase) for query enrichment. However, we found that they are often too coarse-grained, thus bringing limited information for fine-grained rankings.

### 4.3 ToTER with Labeled Data

While ToTER mainly focuses on scenarios without available labeled data from the target corpus, it can also benefit from harnessing $(q, d)$ labels. First, ToTER can directly use $q$ for class relevance learning (Eq.5) by treating it as an additional document. By using the class labels of its relevant document $d$, we can reflect their relevance into the class estimator. Also, the enriched queries by ToTER can be directly used to enhance the fine-tuning of the retriever and reranker. That is, we use $q^{QEP}$ instead of $q$ as model input, where the fine-tuning follows the standard contrastive learning [21].

## 5 EXPERIMENTS

We experiment to answer the following research questions:

**RQ1** How does each strategy of ToTER affect retrieval accuracy?
**RQ2** How does ToTER compare with other techniques that use auxiliary corpus knowledge in terms of retrieval accuracy?
**RQ3** Is ToTER compatible with a variety of PLM-based retrieval models used for each retrieval stage?
**RQ4** How does the labeled data affect the effectiveness of ToTER?
**RQ5** How sensitive is ToTER to the taxonomy quality?

We provide details of setup and implementation in Appendix A.3.

**Dataset.** We simulate two theme-specific applications: (1) academic paper search using SCIDOCS dataset [5], (2) product search in e-commerce using Amazon ESCI dataset [46]. We focus on scenarios with no training labels from the target corpus in §5.1 and analyze the impacts of labeled data in §5.2. For evaluation, we use test labels provided with both datasets. The topical taxonomy is obtained using an existing taxonomy completion approach [28]. We provide data statistics and taxonomy construction steps in Appendix A.3.

**Retrieval setup and Metrics.** Without ToTER, the retrieval process follows the standard (1) retrieval, (2) reranking pipeline (§2.1). ToTER has an added SSA step to reduce the initial search space. Thus, with ToTER, the retrieval process follows (1) SSA, (2) retrieval (using CRM), and (3) reranking (using QEP). We employ various ranking metrics for each retrieval stage. For SSA and the first-stage retrieval, we use Recall (R@K), and for the second-stage reranking, we additionally use NDCG (N@K) and MAP (MAP@K). In our experiments, we set the size of search space adjustment $|\mathcal{D}_q^{SSA}| = 2500$ and the candidate set for reranking $|\mathcal{D}_q| = 100$ [34].

### 5.1 Results and analysis (RQ1,RQ2, and RQ3)

We provide a detailed analysis for each stage of retrieval.

*5.1.1 Initial search space adjustment.* We first assess the effectiveness of SSA in filtering out irrelevant documents from the corpus. As this can be seen as a kind of retrieval with a large retrieval size, we compare various retrieval methods. Note that SSA does not require backbone retrieval models, as it only uses the class estimator.

**Compared methods.** (a) Lexical and sparse model: BM25 [49], docT5query [42], (b) pre-trained PLMs for each corpus: SCIBERT [2] & BERT-base [7][8], (c) unsupervised domain adaptation: GenQ [52], (d) PLM-based retriever: Contriever-MS [17], TAS-B [14], ColBERT [22], SPLADE++ [9]. GenQ uses synthetic data generated for the target corpus[9], and other retrievers in (d) are fine-tuned using massive labeled data from the general domain [40].

**Findings.** Figure 2 presents recalls (R@K) for varying sizes of retrieved documents (K). SSA consistently achieves the highest recall when K is large ($\geq 10^3$), which shows its efficacy in accurately filtering out irrelevant documents. Using class relevance learning, ToTER categorizes documents based on the theme-specific taxonomy. This approach aids in identifying the central subject of documents, which may not be effectively captured by the lexical and semantic similarity based on word overlap and contextual meaning.

Among the competitors, PLM-based retrievers, fine-tuned with vast labeled data, consistently show high recalls. On the other hand, GenQ, fine-tuned with synthetic data, shows limited performance. This result also aligns with [4]. We find that the generated queries are often trivial and fail to reflect the domain-specific knowledge, which may lead to suboptimal results (provided in Appendix A.4.3). Lastly, the effectiveness of SSA rapidly declines when K is small ($\leq 5\times10^2$). This outcome is expected, given that SSA only considers the overlap degree of relevant topic classes.

*5.1.2 Retrieval.* We assess the effectiveness of CRM for the first-stage retrieval. We select three backbone models, which show competitive performance and also represent three different encoding strategies: Contriever-MS, SPLADE++, and ColBERT. These correspond to single-vector, sparse, and multi-vector representation models, respectively. We report results for both (a) without SSA (i.e., retrieval from $\mathcal{D}$) and (b) with SSA (i.e., retrieval from $\mathcal{D}^{SSA}$).

**Compared methods.** We compare CRM with various state-of-the-art methods to improve backbone retrievers using auxiliary corpus knowledge. The first group leverages pseudo-relevance feedback.

- **BERT-QE (BQE)** [65] uses core text segments (or chucks) obtained from top retrieval results to complement the original query.

---

[8] We avoid using pre-trained models that use citation relations [5], as they may reveal the relevance labels of the academic domain.
[9] We fine-tune Contriever-MS using BM25 negatives on the synthetic data.

**Table 2: Retrieval performance comparison on the academic and product domain. Significant differences with the baseline (i.e., retrieval without using ToTER framework) are marked with * (p-value < 0.05 in the one-sample t-test).**

| Search space | Method | Contriever-MS | | | SPLADE++ | | | ColBERT | | |
|---|---|---|---|---|---|---|---|---|---|---|
| | | R@100 | R@500 | R@1000 | R@100 | R@500 | R@1000 | R@100 | R@500 | R@1000 |
| **Academic domain** — Entire corpus ($\mathcal{D}$) | Retriever | 0.3783 | 0.5498 | 0.6216 | 0.3705 | 0.5294 | 0.6004 | 0.3382 | 0.4864 | 0.5624 |
| | w/ BQE | 0.3846 | 0.5543 | 0.6280 | **0.3911** | 0.5523 | 0.6193 | 0.3484 | 0.4991 | 0.5783 |
| | w/ PRF | 0.3815 | 0.5510 | 0.6266 | 0.3852 | 0.5441 | 0.6146 | 0.3484 | 0.5011 | 0.5860 |
| | w/ GAR | **0.3848** | 0.5499 | 0.6218 | 0.3772 | 0.5336 | 0.6033 | 0.3486 | 0.4912 | 0.5629 |
| | w/ LADR | 0.3806 | **0.5626** | **0.6302** | 0.3763 | **0.5577** | **0.6264** | **0.3501** | **0.5339** | **0.6115** |
| | w/ TopicGQA | 0.3835 | 0.5491 | 0.6261 | 0.3672 | 0.5271 | 0.5995 | 0.3393 | 0.4845 | 0.5581 |
| Reduced corpus ($\mathcal{D}^{SSA}$) | Retriever | 0.3887 | 0.5866 | 0.6793 | 0.3913 | 0.5809 | 0.6725 | 0.3586 | 0.5486 | 0.6502 |
| | w/ BQE | 0.3940 | 0.5861 | 0.6836 | 0.4086 | 0.5942 | 0.6856 | 0.3658 | 0.5557 | 0.6553 |
| | w/ PRF | 0.3915 | 0.5862 | 0.6806 | 0.4032 | 0.5896 | 0.6806 | 0.3615 | 0.5547 | 0.6593 |
| | w/ GAR | 0.3928 | 0.5862 | 0.6802 | 0.4099 | 0.5966 | 0.6815 | 0.3964 | 0.5887 | 0.6810 |
| | w/ TopicGQA | 0.3933 | 0.5873 | 0.6802 | 0.3902 | 0.5772 | 0.6719 | 0.3584 | 0.5420 | 0.6382 |
| | w/ ToTER-CRM (ours) | **0.4432*** | **0.6399*** | **0.7268*** | **0.4490*** | **0.6364*** | **0.7252*** | **0.4326*** | **0.6346*** | **0.7232*** |
| **Product domain** — Entire corpus ($\mathcal{D}$) | Retriever | 0.4992 | 0.6962 | 0.8294 | 0.5220 | 0.7129 | 0.7732 | 0.5342 | 0.7091 | 0.7685 |
| | w/ BQE | **0.5256** | 0.7258 | 0.8481 | **0.5582** | 0.7398 | 0.8056 | **0.5603** | **0.7593** | 0.8075 |
| | w/ PRF | 0.5097 | **0.7444** | **0.8572** | 0.5244 | **0.7661** | **0.8383** | 0.5350 | 0.7331 | **0.8265** |
| | w/ GAR | 0.5158 | 0.7231 | 0.8409 | 0.5455 | 0.7214 | 0.8067 | 0.5515 | 0.7421 | 0.8140 |
| | w/ LADR | 0.5157 | 0.7228 | 0.8498 | 0.5377 | 0.7308 | 0.8215 | 0.5435 | 0.7228 | 0.8259 |
| | w/ TopicGQA | 0.5172 | 0.7343 | 0.8298 | 0.5334 | 0.7235 | 0.7924 | 0.5252 | 0.7072 | 0.7818 |
| Reduced corpus ($\mathcal{D}^{SSA}$) | Retriever | 0.5009 | 0.7085 | 0.8555 | 0.5231 | 0.7264 | 0.8229 | 0.5303 | 0.7401 | 0.8394 |
| | w/ BQE | 0.5285 | 0.7411 | 0.8632 | 0.5608 | 0.7467 | 0.8462 | 0.5638 | 0.7602 | 0.8492 |
| | w/ PRF | 0.5124 | 0.7462 | 0.8663 | 0.5311 | 0.7779 | 0.8603 | 0.5355 | 0.7649 | 0.8495 |
| | w/ GAR | 0.5427 | 0.7593 | 0.8564 | 0.5606 | 0.7540 | 0.8416 | 0.5569 | 0.7634 | 0.8396 |
| | w/ TopicGQA | 0.5186 | 0.7381 | 0.8493 | 0.5384 | 0.7286 | 0.8351 | 0.5390 | 0.7459 | 0.8598 |
| | w/ ToTER-CRM (ours) | **0.5515*** | **0.7899*** | **0.8692*** | **0.5717*** | **0.7856*** | **0.8648*** | **0.5661** | **0.7997*** | **0.8625*** |

- **PRF** has been separately studied for the single-vector [59] and multi-vector representation models [55]. They exploit document- and term-level knowledge for query enrichment, respectively. We apply the corresponding PRF method for each backbone model.

The second group uses inter-document similarity via a corpus graph.

- **GAR** [34] uses nearest neighbors in the graph to refine the initial ranking results. It is proposed for the reranking, but we apply it to the retrieval as well, as it brings consistent improvements.
- **LADR** [26] uses lexical retriever in conjunction with the corpus graph to gradually expand the search space for PLM-based retriever. We use LADR-adaptive with no time constraints. As LADR controls the search space, we apply it solely to $\mathcal{D}$.

As discussed in §3, using topic knowledge for PLM-based retrievers has not been studied well. Following a recent approach that uses generative augmentation [35, 37], we devise a new baseline that leverages topic knowledge discovered by PLMs.

- **TopicGQA** uses generative query augmentation [35, 37]. Given a query, we extract its topic using PLMs, and enrich it by adding the predicted topics using the same template to ToTER (Eq.7). We use T0-3B with the prompt proposed in [35] (Appendix A.3.3).

**Findings.** Table 2 presents the retrieval results. First, we observe that retrieval from the filtered corpus via SSA ($\mathcal{D}^{SSA}$) consistently yields higher recalls than retrieval from the entire corpus ($\mathcal{D}$), which again shows the effectiveness of our topic-based SSA. Second, methods that utilize auxiliary corpus knowledge consistently boost the retrieval performance. While BQE and PRF excel in the product domain, corpus graph knowledge demonstrates superior effectiveness in the academic domain. Conversely, TopicGQA, which leverages topic knowledge extracted using PLMs, shows limited effectiveness and even degrades the performance (e.g., R@1000, ColBERT in the academic domain). We notice that TopicGQA often fails to generate contexts reflecting domain knowledge. For example, in Table 1(a), it generates topics like "data ownership" and "prove", while relevant they do not reveal the high-level contexts

of the academic paper. Lastly, CRM consistently shows the highest recall in all setups. Based on the taxonomy, it can identify topic classes reflecting domain knowledge (e.g., "cryptography", "computer security"). The topical relatedness is incorporated with the semantic similarity, providing complementary knowledge to each other. These observations collectively show the effectiveness of using corpus taxonomy for theme-specific retrieval.

*5.1.3 Reranking.* We assess the effectiveness of QEP for the second-stage reranking. Following [34, 52], we use two backbone models: MiniLM-L-12 [54], MonoT5-base [41]. We report results for reranking top-100 results from both (a) retriever and (b) SSA & CRM.[10]

**Compared methods.** We use GAR [34], the state-of-the-art method proposed for the reranking stage, as our main competitor. We also compare TopicGQA. Note that QEP and TopicGQA only differ in the way of generating contexts to enrich queries.

**Findings.** In Table 3, similar to results in Table 2, TopicGQA shows limited performance. For the query in Table 1(b), it generates contexts of "dye", "peroxide", and "ammonia", failing to add new information. The best performance is consistently achieved by using all three strategies of ToTER. QEP differs from TopicGQA in that it identifies core phrases using both the topic class knowledge and top-ranked results, under the idea of relevance model [27]. These processes are guided by the taxonomy reflecting user-interested aspects, which may not be effectively revealed from the corpus graph. Based on the findings in §5.1, we conclude that each strategy of ToTER effectively enhances retrieval in each stage (RQ1, RQ2) and also has great compatibility with PLM-based models (RQ3).

## 5.2 Study of ToTER (RQ4, RQ5)

We provide **ablation, hyperparameter, and case study in the Appendix A.4**. Here, we report the results with Contriever-MS and MiniLM-L-12 on the product domain which has training labels.

---

[10]We use the retrieval results of Contriever-MS (for the academic domain) and SPLADE++ (for the product domain), which show the highest recalls within top-100.

**Table 3: Reranking performance comparison on the academic and product domain. Significant differences with the baseline (i.e., reranking without using ToTER framework) are marked with * (p-value < 0.05 in the one-sample t-test).**

| Candidate set generation | Method | MiniLM-L-12 | | | | | MonoT5-base | | | | |
|---|---|---|---|---|---|---|---|---|---|---|---|
| | | N@3 | N@10 | MAP@10 | R@10 | R@50 | N@3 | N@10 | MAP@10 | R@10 | R@50 |
| **Academic domain** | | | | | | | | | | | |
| Retriever | no reranking | 0.1589 | 0.1652 | 0.0966 | 0.1726 | 0.3166 | 0.1589 | 0.1652 | 0.0966 | 0.1726 | 0.3166 |
| | Reranker | 0.1695 | 0.1760 | 0.1030 | 0.1827 | 0.3347 | 0.1748 | 0.1835 | 0.1078 | 0.1936 | **0.3368** |
| | w/ GAR | **0.1701** | **0.1767** | **0.1036** | **0.1841** | **0.3358** | **0.1752** | **0.1854** | **0.1093** | **0.1972** | **0.3368** |
| | w/ TopicGQA | 0.1666 | 0.1753 | 0.1023 | 0.1822 | 0.3301 | 0.1727 | 0.1800 | 0.1060 | 0.1876 | 0.3309 |
| Retriever w/ ToTER-SSA, CRM | no reranking | 0.1748 | 0.1838 | 0.1074 | 0.1949 | 0.3633 | 0.1748 | 0.1838 | 0.1074 | 0.1949 | 0.3633 |
| | Reranker | 0.1780 | 0.1852 | 0.1090 | 0.1953 | 0.3634 | 0.1758 | 0.1868 | 0.1090 | 0.1997 | 0.3663 |
| | w/ GAR | 0.1784 | 0.1870 | 0.1101 | 0.1979 | 0.3633 | 0.1794 | 0.1900 | 0.1118 | 0.2013 | 0.3671 |
| | w/ TopicGQA | 0.1719 | 0.1829 | 0.1072 | 0.1918 | 0.3632 | 0.1752 | 0.1854 | 0.1087 | 0.1962 | 0.3634 |
| | w/ ToTER-QEP (ours) | **0.1821*** | **0.1915*** | **0.1126*** | **0.2026*** | **0.3660*** | 0.1828 | **0.1930*** | 0.1137 | 0.2048 | **0.3732*** |
| **Product domain** | | | | | | | | | | | |
| Retriever | no reranking | 0.2917 | 0.2845 | 0.1592 | 0.2401 | 0.4425 | 0.2917 | 0.2845 | 0.1592 | 0.2401 | 0.4425 |
| | Reranker | **0.2972** | 0.2937 | 0.1664 | 0.2513 | 0.4544 | **0.3317** | 0.3214 | 0.1883 | **0.2642** | 0.4965 |
| | w/ GAR | **0.2972** | **0.2986** | **0.1697** | **0.2610** | **0.4741** | **0.3317** | **0.3217** | **0.1892** | **0.2642** | **0.5061** |
| | w/ TopicGQA | 0.2965 | 0.2952 | 0.1658 | 0.2522 | 0.4621 | 0.3205 | 0.3043 | 0.1720 | 0.2621 | 0.4967 |
| Retriever w/ ToTER-SSA, CRM | no reranking | 0.3048 | 0.2856 | 0.1603 | 0.2459 | 0.4627 | 0.3048 | 0.2856 | 0.1603 | 0.2459 | 0.4627 |
| | Reranker | 0.2903 | 0.2942 | 0.1644 | 0.2601 | 0.4689 | 0.3317 | 0.3215 | 0.1889 | 0.2642 | 0.5048 |
| | w/ GAR | 0.3047 | 0.3021 | 0.1714 | 0.2654 | 0.4759 | 0.3318 | 0.3253 | 0.1916 | 0.2699 | 0.5098 |
| | w/ TopicGQA | 0.2978 | 0.3039 | 0.1734 | 0.2608 | 0.4785 | 0.3301 | 0.3185 | 0.1887 | 0.2626 | 0.5021 |
| | w/ ToTER-QEP (ours) | **0.3189*** | **0.3139*** | **0.1818*** | **0.2701*** | **0.4891*** | **0.3416** | **0.3304*** | **0.1921** | **0.2729*** | **0.5227*** |

### 5.2.1 ToTER with labeled data.

Table 4 presents the performance of ToTER with varying amounts of labeled data. L0/L1/L2/L3 denote the setups using 0/33/66/100% of the available training labels, respectively. As discussed in §4.3, ToTER can directly use labeled data for its training process. We observe that the overall performance of the retriever and reranker is largely improved by fine-tuning with labeled data. Next, we observe that ToTER effectively enhances the retrieval process in all setups. In specific, SSA effectively narrows down the initial search space without hurting recalls of the fine-tuned retriever. Furthermore, CRM and QEP consistently improve both the retriever and reranker. These results show that ToTER can effectively leverage the labeled data, yielding a good synergy with the fine-tuned PLM-based models (RQ4).

### 5.2.2 Impacts of taxonomy quality.

The power of ToTER is primarily attributed to the topical taxonomy. While the methodology for taxonomy completion has been extensively studied and well-established, it's crucial to assess the robustness of ToTER regarding the quality of the given taxonomy. To this end, we consider two aspects measuring taxonomy quality [28]: (1) topic completeness, which assesses how fully the topic nodes cover the true topics, and (2) term coherence, which assesses the semantic relatedness of terms (or phrases) within a topic node. To impair completeness, we apply *random pruning*, which randomly removes a node and all its child nodes.[11] To impair coherence, we apply *level-wise node shuffling*, which randomly swaps nodes at the same level.[12] Note that such random shuffling corresponds to an extreme scenario.

In Figure 3, we observe that although both types of noise degrade the effectiveness, ToTER has a considerable degree of robustness. In particular, it shows a more stable performance for the pruning. We conjecture this stability arises because the relevance to the removed nodes can be partially inferred from the relevance to the remaining topic nodes. For the node shuffling, noise at deeper levels has higher impacts. This can be due to the increased node numbers at deeper levels and the decreased number of child nodes which can help to reduce the impacts of shuffling. Based on the observations, we conclude that ToTER has a certain degree of robustness to taxonomy

---

[11]We apply pruning by controlling the ratio of removed nodes to the total nodes.
[12]The root node has level 0. We set the shuffling ratio as 10%.

---

**Table 4: Performance of ToTER with labeled data. Significant differences with the baseline are marked with * (p-value < 0.05 in the one-sample (L0) /paired (L1-L3) t-test).**

| | Method | L0 | L1 | L2 | L3 |
|---|---|---|---|---|---|
| **R@100** | Retriever | 0.4992 | 0.5168 | 0.5179 | 0.5527 |
| | w/ ToTER-SSA | 0.5009 | 0.5191 | 0.5202 | 0.5665 |
| | w/ ToTER-SSA, CRM | **0.5515*** | **0.5502*** | **0.5865*** | **0.6098*** |
| **R@500** | Retriever | 0.6962 | 0.7592 | 0.7801 | 0.8071 |
| | w/ ToTER-SSA | 0.7085 | 0.7697 | 0.7802 | 0.8090 |
| | w/ ToTER-SSA, CRM | **0.7899*** | **0.7823*** | **0.8179*** | **0.8434*** |
| **R@1K** | Retriever | 0.8294 | 0.8333 | 0.8442 | 0.8714 |
| | w/ ToTER-SSA | 0.8555 | 0.8353 | 0.8468 | 0.8718 |
| | w/ ToTER-SSA, CRM | **0.8692*** | **0.8636*** | **0.8671*** | **0.8906*** |

| L3 | N@3 | N@10 | MAP@10 | R@10 | R@50 |
|---|---|---|---|---|---|
| Retriever & Reranker | 0.3188 | 0.3107 | 0.1746 | 0.2616 | 0.5072 |
| w/ ToTER | **0.3241** | **0.3219** | **0.1907*** | **0.2778*** | **0.5249*** |

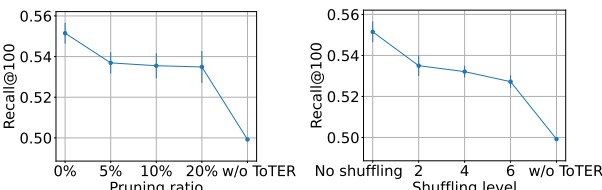

**Figure 3: Retrieval results with taxonomy impaired in terms of (left) topic completeness and (right) term coherence.**

quality and can effectively enhance retrieval using the existing taxonomy completion methods (RQ5).

## 6 CONCLUSION

We propose a new ToTER framework to enhance PLM-based retrieval in theme-specific applications using a corpus topical taxonomy. ToTER identifies the central topics of queries and documents with the guidance of topical taxonomy via class relevance learning, and exploits their topical relatedness to complement semantic matching by PLM-based models. ToTER introduces three strategies, SSA, CRM, and QEP, which gradually focus on fine-grained ranking following the retrieve-then-rerank pipeline. Our comprehensive experiments on two real-world datasets ascertain the benefits of using topical taxonomy and demonstrate the effectiveness of ToTER.

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

# A  APPENDIX

The code and data will be released via the author's repository upon acceptance.

## A.1  Related work for topic mining and taxonomy completion

**Topic Mining.** Traditional topic models like pLSA [13] and LDA [3] discover semantically relevant topics from the corpus in an unsupervised manner. While they are effective, they are unable to incorporate supervision which can provide valuable knowledge for user-interested aspects of the application. Subsequent supervised topic models like Labeled LDA [45] and SSHLDA [36] have been proposed to incorporate such supervision. However, they require massive topic-labeled documents, which are difficult to obtain in practice [38]. To reduce reliance on labels, several studies have focused on weakly supervised models that use a set of user-interested seeds for topic discovery. Seeded LDA [18] biases the generative process of the regular topic model by introducing a seed topic distribution, and CatE [38] uses category names to learn embedding space that reflects distinctiveness among seeds. However, they produce a flat topic structure of the corpus and thus do not reveal hierarchical relations of the topics [15].

**Topical taxonomy completion.** Topical taxonomy represents the latent topic hierarchy of document collections, providing valuable knowledge of contents in many applications. Early methods build a corpus topic taxonomy from scratch by extracting discriminative term clusters from the corpus in a hierarchical fashion [61]. To generate user-interested topic structure, recent approaches [15, 28, 29, 50] have started with a seed taxonomy rooted in human knowledge of the application and expanded it by discovering novel topics from the target corpus. The seed taxonomy is used to guide the whole process of discovering new topics and expanding the taxonomy. Specifically, [15] trains classifiers to capture user-interested relations from parent-child topic pairs, [28] recursively clusters phrases to identify new subtopics based on the known topic relations. Very recently, [29] generates topic-conditioned terms by leveraging hierarchical relations from the seed taxonomy. ToTER can use any topical taxonomy completion method, we use TaxoCom [28] for our experiments. Please refer to Appendix A.3.2 for details.

## A.2  Training details of ToTER

The training process of ToTER is provided in Algorithm 1. We first initialize the class relevance estimator $g$, which has the training parameters for the class encoder (i.e., $\theta_{GNN}$) and bilinear interaction matrix (i.e., $\mathbf{M}$). Then, we generate the silver labels $\mathbf{y}_d^s$ and use them to warm up $g$. After the warm-up, we train $g$ with the collective knowledge distillation. In specific, for each document $d$, we retrieve a set of similar documents $\mathcal{D}_d$ with CRM, and generate the collective labels $\mathbf{y}_d^c$ by averaging their class relevance distributions. In practice, we update the collective topic labels $\mathbf{y}_d^c$ every $t$ epochs, which makes the training process more efficient and robust. In this work, we set $|\mathcal{D}_d| = 10$ and $t = 25$.

---

**Algorithm 1:** Training algorithm of ToTER.

---

**Input** : A target corpus $\mathcal{D}$, a corpus topical taxonomy $\mathcal{T}$, a pre-trained language model $P$, a retriever $f$, a update period $t$

**Output**: Trained class relevance estimator $g$

1  Randomly initialize the training parameters of $g$
2  Generate silver labels $\mathbf{y}_d^s$ for all $d \in \mathcal{D}$ using $\mathcal{T}$ and $P$   // §4.1.1
3  Warm-up the estimator $g$ using only $\mathbf{y}_d^s$   // Eq.5

   /* Collective knowledge distillation   */
4  **for** $e = 1, ..., epoch_{max}$ **do**
5    **foreach** $d \in \mathcal{D}$ **do**
6      **if** $e \% t == 0$ **then**
           /* Collective label generation   */
7        Retrieve $\mathcal{D}_d$ using $f$ with CRM   // §4.2.2
8        Compute class relevance as $\{\hat{\mathbf{y}}_{d'} \mid d' \in \mathcal{D}_d\}$ using $g$
9        Obtain collective labels $\mathbf{y}_d^c = \text{AVG}(\{\hat{\mathbf{y}}_{d'} \mid d' \in \mathcal{D}_d\})$
10     Train the estimator $g$ using $\mathbf{y}_d^c$   // Eq.5

---

## A.3  Experiment details

*A.3.1  Dataset.* In our experiment, we simulate two theme-specific applications: (1) academic paper search using SCIDOCS dataset [5, 52], (2) product search in e-commerce using Amazon ESCI dataset [46].[13] SCIDOCS dataset is widely used as a benchmark dataset evaluating the zero-shot prediction capability of retrieval models [52]. Amazon ESCI dataset is adopted from KDD Cup 2022-Task 1: Query-Product Ranking. We use the English (US) data and treat 'E (exact match)' as the relevant relation. We evaluate the effectiveness of each method in ranking relevant documents (products) above all non-relevant ones. Each document contains the product title, product description, and product bullet points. For evaluation, we use the test labels provided with both datasets. Table 5 summarizes the data statistics.

**Table 5: Data statistics of two theme-specific domain datasets. Avg. D/Q indicates the average number of relevant documents per query.**

|  | Academic domain | Product domain |
|---|---|---|
| #Corpus | 25,657 | 601,354 |
| #Training query | - | 20,888 |
| #Test query | 1,000 | 8,956 |
| Avg. D/Q | 4.9 | 8.83 |

*A.3.2  Corpus Topical taxonomy construction.* The corpus topical taxonomy is obtained by applying the existing taxonomy completion method [28] on a seed taxonomy. For the seed taxonomy, we utilize the fields of study hierarchy from Microsoft Academic Graph [51] (for the academic domain) and Amazon store taxonomy[14] sourced from Amazon.com (for the product domain). Each seed taxonomy mirrors user interest in each application. The former reflects researchers' inclination towards structuring academic concepts and studies, while the latter embodies customers' interest in browsing and selecting products.

---

[13]https://github.com/amazon-science/esci-data
[14]https://www.amazonlistingservice.com/blog/amazon-store-taxonomy-organization

Based on the seed taxonomy, we conduct taxonomy completion which completes and adjusts the taxonomy for the target corpus. This is a critical step to ensure the taxonomy aligns with the target corpus. Note that the seed taxonomy is incomplete; it contains numerous topics irrelevant to the corpus as well as failing to cover all topics. For example, in the case of the SCIDOCS dataset, we discovered that over 95% of topic classes from the seed taxonomy have no (or a very weak) relevance to documents in the corpus. Also, as it does not cover all specific topics in the corpus, we need to expand it by identifying new topics not present in the original seed taxonomy. We use the recently proposed TaxoCom [28] to obtain the corpus topical taxonomy. We use the official implementation provided by the authors.[15] A notable change from the original implementation is that we additionally use PLM knowledge for more effective topic discovery [64]. Table 6 provides the statistics of the constructed taxonomies.

**Table 6: Taxonomy statistics of two theme-specific domain datasets.**

|               | Academic domain | Product domain |
|---------------|-----------------|----------------|
| #Topic classes | 4,028         | 14,954         |
| #Edges        | 8,445           | 18,360         |
| Depth         | 5               | 10             |

*A.3.3 Implementation details.* In our experiments, we use BEIR benchmark framework[16] for evaluating all compared methods. For BM25, we use Elasticsearch. For docT5query and GenQ, we use T5 models with checkpoints provided by BEIR.[17] For all compared PLM-based retrieval and reranking models, we use checkpoints that are publicly available: Contriever-MS[18], TAS-B[19], ColBERT[20], SPLADE++[21], MiniLM-L-12[22], and MonoT5-base[23]. For multi-vector representation models, we use ColBERT.v1 solely for compatibility with the public ColBERT-PRF implementation.

To generate the corpus graph, we use Contriever-MS for the academic domain and SPLADE++ for the product domain, as they consistently show the highest recalls within top-100. The number of neighbors in the graph is set as 10, as it shows stable results in both GAR [34] and LADR [26]. For BERT-QE [65], we set both the number of top-ranked documents and the number of core segments (or chunks) as 10, following the paper. For PRF of the single and sparse representation model (i.e., Contriever-MS, SPLADE++), we set the number of documents for query enrichment as 5. For PRF for the multi-vector representation model (i.e., ColBERT), we use the official implementation and provided values. For other baseline-specific hyperparameters, we follow the recommended values in the original papers [26, 34, 55, 59]. For TopicGQA, we use T0-3B[24] with the prompt suggested in [35]: *"Based on the query, generate a bullet-point list of relevant topics present in relevant documents:"*.

---

[15]https://github.com/donalee/taxocom/tree/main
[16]https://github.com/beir-cellar/beir
[17]castorini/doc2query-t5-base-msmarco, BeIR/query-gen-msmarco-t5-base-v1
[18]facebook/contriever-msmarco
[19]msmarco-distilbert-base-tas-b
[20]https://github.com/terrierteam/pyterrier_colbert
[21]naver/splade-cocondenser-ensembledistil
[22]cross-encoder/ms-marco-MiniLM-L-12-v2
[23]castorini/monot5-base-msmarco-10k
[24]bigscience/T0_3B

**Table 7: Ablation results for class relevance learning of training phase.**

|                          | R@100  | R@500  | R@1000 |
|--------------------------|--------|--------|--------|
| ToTER-SSA, CRM           | **0.4432** | **0.6399** | **0.7268** |
| $y^s$ only               | 0.3809 | 0.6080 | 0.7131 |
| CKD → Self-training      | 0.4327 | 0.6242 | 0.7226 |
| w/o ToTER                | 0.3783 | 0.5498 | 0.6216 |

**Table 8: Ablation results for each strategy of inference phase.**

|     |                          | R@2500 | R@5000 | R@10000 |
|-----|--------------------------|--------|--------|---------|
| SSA | ToTER-SSA                | **0.8439** | **0.9053** | **0.9599** |
|     | Low-level focus          | 0.7832 | 0.8784 | 0.9505 |
|     | High-level focus         | 0.5569 | 0.6964 | 0.8552 |

|     |                          | R@100  | R@500  | R@1000 |
|-----|--------------------------|--------|--------|--------|
| CRM | ToTER-SSA, CRM           | **0.4432** | **0.6399** | **0.7268** |
|     | Low-level focus          | 0.3864 | 0.5739 | 0.6623 |
|     | High-level focus         | 0.3361 | 0.4606 | 0.5087 |
|     | w/o ToTER                | 0.3783 | 0.5498 | 0.6216 |

|     |                          | N@10   | R@10   | R@50   |
|-----|--------------------------|--------|--------|--------|
| QEP | ToTER-SSA, CRM, QEP      | **0.1915** | **0.2026** | **0.3660** |
|     | w/o top-ranked docs.     | 0.1759 | 0.1855 | 0.3549 |
|     | w/o ToTER                | 0.1760 | 0.1827 | 0.3347 |

For fine-tuning with labeled data in the product domain (§5.2.1), we use sentence_transformers framework [47]. We continue fine-tuning for 10 epochs with a learning rate of $7e^{-5}$. We discovered that further increasing the training epochs consistently degrades the retrieval accuracy. For ToTER, we set the retaining percent $m = 10\%$, the number of core phrases $k = 5$. We provide a sensitivity study with varying $m$ and $k$ in A.4.2. Following BERT-QE [65], the number of top-ranked documents (for collective labels and QEP) is set to 10. Lastly, the number of GNN layers ($L$) in the class relevance estimator is set to 1, but it can be further tuned.

## A.4 Supplementary results

In this section, we report the results with Contriever-MS (for retriever) and MiniLM-L-12 (for reranker).

*A.4.1 Ablation study.* We provide a performance comparison with alternative design choices in the academic domain. Table 7 provides ablation results for the training phase of ToTER (i.e., class relevance learning), and Table 8 provides ablation results for the inference phase of ToTER.

For the training phase, we compare two ablations intended to verify the effectiveness of our silver label generation and collective topic knowledge distillation (CKD): **(1) $y^s$ only** solely uses the generated silver labels for the class relevance learning without CKD, and **(2) CKD → Self-training** replaces CKD with the standard self-training. The self-training is a well-established semi-supervised learning technique used to achieve better generalization when the given labels are incomplete [57]. The core difference between self-training and CKD is that self-training generates additional training signals using model prediction on individual data instances, whereas CKD uses collective knowledge of the averaged prediction on semantically similar documents. Note that the taxonomy is corpus-level knowledge and ground-truth labels for classification (i.e., topic class labels for each document) are not available. For this

reason, we analyze the effectiveness of the ablations in terms of the retrieval performance.

From Table 7, we observe that both silver labels and CKD play important roles in class relevance learning. First, the class relevance estimator only trained with $\mathbf{y}^s$ consistently improves the retrieval performance. This supports the effectiveness of our silver label generation strategy. Also, we observe that exploring relevant but unlabeled classes is highly important. Both self-training and CKD bring considerable improvements compared to using only the silver label. However, replacing CKD with self-training consistently degrades the retrieval effectiveness. We also find that self-training is rather unstable compared to using CKD.

For the inference phase, we compare ablations for each strategy: SSA, CRM, and QEP. For SSA and CRM, which utilize the estimated class relevance distributions, we compare two alternative design choices: **(1) low-level focus** emphasizes relevance to the more specific, narrower classes found at the lowermost levels, specifically the lowest two levels. **(2) high-level focus** is the opposite choice of the low-level focus. It targets more general, broader classes closer to the root node, specifically the top three levels. For QEP, we compare an ablation that ablates the use of top-ranked documents in the core phrase identification, denoted as **(3) w/o top-ranked docs.** Specifically, from the set of relevant class phrases $\{P_j | \hat{b}_{qj} = 1\}$, we collect the top-$k$ core phrases $P^q$ that most frequently appear across the corpus.

From Table 8, we observe that both low-level and high-level focus results in suboptimal results, and considering both board and specific topics consistently leads to the best recalls. In particular, ignoring low-level classes (i.e., high-level focus) more drastically degrades the retrieval accuracy. Also, we observe that using the frequency information from top-ranked documents is indeed effective in finding proper contexts to enrich queries. For example, for the query in Table 1(a), **w/o top-ranked docs** identifies core phrases like "key", "scale", "software", "management", and "system", which are relevant but not closely related to the query compared to the phrases obtained from ToTER. These results support the validity of our design choice that uses the overall topic class distributions for SSA and CRM, and our core phrase identification based on the relevance model for QEP.

*A.4.2 Hyperparameter study.* We provide analyses to guide the hyperparameter selection of ToTER. Specifically, we investigate the effects of two hyperparameters introduced by ToTER: (1) $m$ denotes the retaining percent of topic classes (§4.2), and (2) $k$ denotes the number of core phrases used to enrich queries in QEP (§4.2.3). We report the results in the academic domain. Similar tendencies are observed in the product domain. First, Table 9 presents the effects of $m$ on retrieval. We observe stable performance overall with $m$ around $5 - 20\%$. In this work, we set $m = 10\%$. Second, Table 10 presents the effects of $k$ on reranking. We observe the best performance is achieved with $k$ around $5 - 7$. In this work, we set $k = 5$.

*A.4.3 Additional case studies.* We provide additional case studies of ToTER and generation-based baselines (i.e., GenQ, TopicGQA). Table 11 presents the generated contexts from GenQ and TopicGQA for examples in Table 1. We observe that the generated queries

**Table 9: Retrieval performance with varying retaining percent $m$.**

| $m$ (%) | Recall@100 | Recall@500 | Recall@1000 |
|---|---|---|---|
| 5 | 0.4396 | 0.6353 | 0.7216 |
| 10 | **0.4432** | 0.6399 | 0.7268 |
| 15 | 0.4414 | **0.6402** | 0.7274 |
| 20 | 0.4410 | 0.6401 | **0.7278** |

**Table 10: Reranking performance with varying sizes of the number of core phrases $k$.**

| $k$ | NDCG@10 | Recall@10 | Recall@50 |
|---|---|---|---|
| 2 | 0.1779 | 0.1888 | 0.3550 |
| 5 | 0.1915 | 0.2026 | **0.3660** |
| 7 | **0.1927** | **0.2044** | 0.3633 |
| 10 | 0.1866 | 0.1956 | 0.3632 |
| w/o ToTER | 0.1760 | 0.1827 | 0.3347 |

and topics do not effectively reveal the domain-specific contexts. These results again show the difficulty of retrieval in theme-specific applications and the importance of proper use of well-structured corpus knowledge.

Lastly, Table 12 presents additional case studies for each domain. Similar to the previous case studies in Table 1, ToTER can effectively identify common topic classes and core topic phrases, guided by the theme-specific taxonomy. This information complements the semantic matching by PLM-based retrieval models, allowing the models to more accurately find relevant documents.

**Table 11: Case studies of generation-based baselines for examples in Table 1. For GenQ for the product domain, we present results of Document A.**

| | (a) Academic domain | (b) Product domain |
|---|---|---|
| GenQ | "what are pors." 

 "what is por." | "what is the name for the hair dyes." 
 "what is onc hair color." |
| TopicGQA | data ownership, prove, store, proving, possession | dye, peroxide, ammonia, hair, natural |

**Table 12: Additional case studies of ToTER. We use Contriever-MS (retriever) and MiniLM-L-12 (reranker). Contents closely related to the query are denoted in bold.**

| (a) Academic domain | | (b) Product domain | |
|---|---|---|---|
| Query | Exploring venue popularity in foursquare | Query | 0.07 lash extensions not easy fans |
| Document A (relevant, rank: top-167) | **Event-based social networks**: linking the **online and offline social worlds**. Newly emerged event-based online **social services**, such as Meetup and Plancast, have experienced increased **popularity** and rapid growth. From these services, we observed a new type of social network - event-based social network (EBSN). ... This paper is the first research to study EBSNs at scale and paves the way for future studies on this new type of social network. | Document A (relevant, rank: top-57) | Volume **Lash Extensions 0.07** D Curl Mix 8-15mm Eyelash Extensions. ... One Second easily grafted **eyelash**, a special craft that allowing you to create 2D10D **fans**. Thickness 0.03/0.05/**0.07**/0.10mm. CurlC/CC/D/DD curl **lash extensions**. Single length 8mm25mm. Mixed length 8-15mm MIX, 9-16mm MIX, 15-20mm MIX in one tray. The root of the volume **lashes extensions** will not separate, any flowering and novices can operate. ... |
| ToTER rank: top-28 | Common topic classes: diffusion, social network, social behavior, world wide web, computer science, economy, economics | ToTER rank: top-7 | Common topic classes: makeup brushes and tools, false lashes, tools and accessories, eye, beauty and personal care |
| | Core phrases: point of interest, location, check, social network, foursquare | | Core phrases: eyelash, fan, extensions, volume lash extensions, lash |

