# OpenReview forum: "Improving Retrieval in Theme-specific Applications using a Corpus Topical Taxonomy"
_ACM.org/TheWebConf/2024/Conference — TheWebConf24_

### Official Review · Reviewer_JV2Q · 2023-11-16

**Novelty:** 4
**Technical Quality:** 4

**Review:**

Summary
The paper introduces a framework named ToTER (Topical Taxonomy Enhanced Retrieval), designed to address limitations in applying PLMs (Pretrained Language Models) to theme-specific contexts. This framework aims to capture theme-specific information to improve retrieval effectiveness. Specifically, during the training phase, it constructs a topical taxonomy for the target corpus by combining an existing taxonomy with the corpus using a silver labels generation strategy and CKD (Contrastive Knowledge Distillation). In the inference phase, ToTER uses the corpus's topical taxonomy to guide the retrieval of documents relevant to the query's theme, employing three strategies: SSA (Sub-Space Alignment), CRM (Contextual Relevance Matching), and QEP (Query Expansion Processing). As a plug-and-play solution, ToTER can be flexibly applied to enhance various PLM-based retrievers. The authors validate ToTER's effectiveness through comprehensive experiments on two real-world datasets, demonstrating the benefits of using a topical taxonomy.

Strengths:
1.The use of Topical Taxonomy to enhance PLM-based retrieval capability is both effective and innovative. It reduces the retrieval space by pruning the Topical Taxonomy tree.
2.The paper is well-written, accurately using terms related to themes, topics, relevance, and similarity.

Weaknesses:
1.In some tables, like Table 1, the meanings of the attributes and their relationships are ambiguous, and no clarification is provided.
2. The work lacks novelty, appearing to be an incremental development based on others. The Topical Taxonomy is constructed using a method from another work named TaxoCom.

**Questions:**

1.Regarding line 342, why is a filtering step necessary after assigning labels to every document? Is it because each document can be assigned multiple labels? How is the number of reliable labels determined, or are they assigned based on empirical verification?

2.Could you provide more details about the role of topical phrases in GCNs, as mentioned in line 438? Specifically, how is hierarchical information related to themes represented, and what specific information should be focused on? An intuitive diagram illustrating the construction of the ego diagram would be helpful.

3.On line 454, what distinguishes a topic label from a silver label, and could you explain the significance of y_d^c?

4.In line 461, given the additional components added for retrieval and reranking, could you elaborate on the time and space complexity during training and inference?

5.What impact does the prompt template have, as mentioned in line 592?

6.In line 611, the absence of ablation studies discussing the effectiveness of each proposed strategy within your framework is noted. It would be beneficial to understand their specific roles and experimental impacts. Additionally, in Table 2, the experiments for ToTER-CRM (ours) seem absent in the Entire Corpus section, similar to the retriever part in Table 3. Could you focus on highlighting the most novel or core aspects of your work, rather than incremental additions?

**Ethics Review Description:**

Null

**Reviewer Confidence:**

4: The reviewer is certain that the evaluation is correct and very familiar with the relevant literature

**Scope:**

3: The work is somewhat relevant to the Web and to the track, and is of narrow interest to a sub-community

---

### Official Review · Reviewer_JVqw · 2023-11-17

**Novelty:** 5
**Technical Quality:** 7

**Review:**

The paper introduces an approach, ToTER, for using domain specific taxonomies for retrieval in specialized domains such as scientific text and product search. ToTER consists of two stages: 1) learning a multi class classification model based on weak supervision, followed by inferring labels for the corpus, and 2) using the topic enhanced documents in the corpus for a taxonomy enhanced retrieval. The taxonomy enhanced documents are used in multiple ways: 1) a binary vector search for reducing the number of documents for subsequent ranking stages, 2) use of the topic vectors for a first stage ranking, 3) crossencoder re-ranking with a query enriched using pseudo relevance feedback. The paper compares the proposed approach to an extensive number of baselines and analyzes various aspects of ToTER. The approach sees consistent improvements over baselines.

Pros:
- The paper is largely well-written.
- The papers experiments are thorough and convincing.
- The presented idea is interesting and combines existing pretrained LM based retrieval with a more symbolic approach well. It is nice to see that this performs well, it has the potential to be useful in domain specific retrieval scenarios.

Cons (elaborated below):
- There are several hyperparameters to the proposed method and its not clear to what extent the proposed method is reliant on those hyperparameters - it is also not obvious how one should set these values.
- While the paper is extensive in discussing ToTER it could benefit from discussing the inference costs of ToTER.

**Questions:**

- What is the motivation behind the formulation of Eq 4? What is the rationale behind setting \rho=0.1?
- Since it seems like the performance of ToTER is reliant on the topic classifier, did any experiments evaluate the this model? Please consider reporting this in the appendix -- even if the results are with some kind of silver labels.
- Did any experiments explore the specific formulation of the classifier? How was the BERT+GCN approach arrived at? Were simpler formulations tried? I'm wondering if the effectiveness of the retrieval can be maintained with a simpler classifier which may underperform your current classifier.
- Sec 4.2: "denotes that d has a certain degree of relevance to c_j" -- in deciding of a topic class is relevant to a document did the experiments use a threshold? How was the threshold set?
- Sec 4.2: Please describe how large the taxonomies used in the experiments are? Consequently, how large of a vector is used for representing the documents and queries?
- Sec 4.2: Do I understand correctly that to perform retrieval even the query needs to get mapped to a binary vector? Is it correct to think that as the size of the taxonomy (depth, width) increases this process gets more expensive? Please discuss these aspects in the paper.
- Sec 4.2.2: Please consider renaming your CRM approach because it seems like CRM refers to the combine(dense retrieval, CRM) as well as the similarity formulated in the second line of Eq 6 resulting in CRM=dense+CRM. This is confusing.
- Sec 4.2.3: Do the experiments use pre-trained crossencoders? What is the training data for the crossencoders?
- Figure 2: Do I understand correctly that the academic and product domains both have two plots each because the number of baselines being compared is too large? If so, please make the y-axis of both plots to lie on the same range. If not, please explain why there are two plots per dataset.

**Reviewer Confidence:**

4: The reviewer is certain that the evaluation is correct and very familiar with the relevant literature

**Scope:**

4: The work is relevant to the Web and to the track, and is of broad interest to the community

---

### Official Review · Reviewer_c2hd · 2023-11-22

**Novelty:** 4
**Technical Quality:** 5

**Review:**

This paper presents an approaches to exploiting taxonomic information to improve retrieval for "theme specific" domains. Theme specific domains is not precisely defined, but seems to imply any domain with a narrow/niche focus. Given a topic specific domain, the authors propose an approach to improve document classification into that taxonomy which, roughly speaking propagates lables from similar documents. Then, these labels are used to improve retrieval at 3 stages: (i) search space filtering using binary operations, (ii) combining topical similarithy with semantic similarity at the initial retrieval stage, and (iii) improving the final ranking phase using a variation of query expansion based on pseudo relevance feedback.

Overall, vertifical search in specialised domains is an important area, and the results in this paper do appear to be promising. I consider this paper to be boderline between accept/reject. While I find the approach to be promising, I have a number of concerns that lead me to be skeptical of the results.

   - Firstly, the novelty of the work is not particularly high. The main novelty seems to be in how the the document lables are refined. The methods for using these labels for improving retrieval are not particularly new or novel. As such, I'm surprised that the authors report such strong improvements over the baselines. I find it hard to believe that adding topical similarity in addition to semantic similarity can lead to the massive improvements reported in Table 2, for example.  The improvements for the final ranker in Table 3 are more modest, but I still find it hard to attribute these improvements to simple query expansion/enrichment. In general, more analysis and experimentaion would help to make these results more convincing.  (EDIT: based on the author response, given the size of the taxonomy used, it now seems more feasible to me that the approach could achieve these improvements).
   - One major omission is that the author never explain how queries are classified. Are they are treated equivalently to "documents" during the label propagation phase? If so, I would argue that this greatly cheapens the results, since an ad hoc retrieval system typically should not need to have advance knowledge of the input queries. If the queries are not processed equivalently to documents, on the other hand, then I have no idea about how the queries are classified. Given that this is core to the paper, this is a serious omission.
   - No details of the taxonomy generation/completion are given in the main paper (and limited details are given in the appendix). Given that the taxonomy is core to the contribution, it's important to understand how it is generated, in particular so that we can understand if there is any potential bias in the process. For example, were queries included in the corpus that was used for taxonomy completion? If so, similar to my point above, this would invalidate many of the conclusions of the work, in my opinion.
  - The paper doesn't feel very self contained, with many important details (taxonomy completion, related work, important experiment details, etc ) place in the Appendix.

**Questions:**

- How are queries classified into the taxonomy?

- It would be great to have more details of the taxonomy completion. Were queries also used as input "documents" at this stage.

**Ethics Review Description:**

No ethics concerns

**Reviewer Confidence:**

3: The reviewer is confident but not certain that the evaluation is correct

**Scope:**

4: The work is relevant to the Web and to the track, and is of broad interest to the community

---

### Official Review · Reviewer_uXEK · 2023-11-23

**Novelty:** 4
**Technical Quality:** 4

**Review:**

1) Approach: The paper introduces an approach, ToTER, for improving retrieval in theme-specific applications. This demonstrates a commitment within the field of information retrieval.

2) Well-Defined Problem: The paper effectively defines the problem of retrieval in theme-specific applications and argues for the importance of well-structured corpus knowledge.

3) Thorough Experiments: The authors conduct experiments using various models and datasets, providing a robust evaluation of the proposed approach.

4) Methodology: The methodology, including taxonomy completion and the use of TaxoCom, is relatively well explained, contributing to the clarity of the paper.

5) Weaknesses/Strengths:
a) Metric Discussion: The paper could provide a more in-depth discussion of the choice of evaluation metrics, explaining why specific metrics were chosen and their implications.
b) Complexity: Some sections contain technical information that may make it difficult for readers without in-depth knowledge of the subject to understand the content.
c) Originality: although the work doesn't seem very original, as the authors show in the state-of-the-art section. However, the work done is substantial and the approach seems to follow a process that combines several sub-approaches to define an information retrieval model emphasizing the importance of theme-specific taxonomy, which is a notable strength (show Figure 1).

**Questions:**

1) Taxonomy Completeness: How did you determine the completeness of the seed taxonomy, and what challenges did you face in this regard?

2) Evaluation Metrics Choice: Could you elaborate on the reasons behind selecting specific evaluation metrics? Are there scenarios where these metrics might not fully capture the system's performance?

3) Generalization: How well does ToTER generalize across different theme-specific domains? Are there specific characteristics of certain domains that might impact its performance?

4) Scalability: How scalable is ToTER, especially in the context of large-scale datasets? Are there computational challenges that might arise with more extensive and diverse corpora?

**Reviewer Confidence:**

3: The reviewer is confident but not certain that the evaluation is correct

**Scope:**

4: The work is relevant to the Web and to the track, and is of broad interest to the community

---

### Decision · Program_Chairs · 2024-01-22

**Decision:**

Accept

**Comment:**

This paper presents an interesting solution towards enhancing retrieval performance by using a theme-specific topical taxonomy under the paradigm of PLM-based ranking. The topic is highly relevant to the search track. The paper is well written and offers extensive experiments. I have skimmed the paper and examined all reviews and rebuttal carefully. Firstly, the idea of uitlizing a topical taxonomy for domain-specific retrieval is promising and will be quite valuable for PLM based ranking methods. Secondly, it is happy to see the currently dominating pre-training is not the only way that we can make further progress, especially on low-resource settings. The construction of a corpus would be cheaper and more flexible for diverse domains. Also, as mentioned by the reviewers, the novelty is not that high, since the utilization of taxonomy is not a new thing. By the way, it is also a natural choice when we need to achieve cost-effective goal. Other comments raised by the reviewers seem to be fixed easily by simple rephrasing. Because the pros far outweigh the cons, I would like to recommend the acceptance given there are sufficient slots.